# MeCP2 nuclear dynamics in live neurons results from low and high affinity chromatin interactions

Francesco M Piccolo[1]*, Zhe Liu[2], Peng Dong[2], Ching-Lung Hsu[2], Elitsa I Stoyanova[1], Anjana Rao[3], Robert Tjian[4], Nathaniel Heintz[1]*

[1]Laboratory of Molecular Biology, Howard Hughes Medical Institute, The Rockefeller University, New York, United States; [2]Janelia Research Campus, Howard Hughes Medical Institute, Ashburn, United States; [3]La Jolla Institute for Allergy and Immunology, La Jolla, United States; [4]Department of Molecular and Cell Biology, Li Ka Shing Center for Biomedical and Health Sciences, CIRM Center of Excellence, University of California, Howard Hughes Medical Institute, Berkeley, United States

**Abstract** Methyl-CpG-binding-Protein 2 (MeCP2) is an abundant nuclear protein highly enriched in neurons. Here we report live-cell single-molecule imaging studies of the kinetic features of mouse MeCP2 at high spatial-temporal resolution. MeCP2 displays dynamic features that are distinct from both highly mobile transcription factors and immobile histones. Stable binding of MeCP2 in living neurons requires its methyl-binding domain and is sensitive to DNA modification levels. Diffusion of unbound MeCP2 is strongly constrained by weak, transient interactions mediated primarily by its AT-hook domains, and varies with the level of chromatin compaction and cell type. These findings extend previous studies of the role of the MeCP2 MBD in high affinity DNA binding to living neurons, and identify a new role for its AT-hooks domains as critical determinants of its kinetic behavior. They suggest that limited nuclear diffusion of MeCP2 in live neurons contributes to its local impact on chromatin structure and gene expression.

*For correspondence:
fpiccolo@rockefeller.edu (FMP);
heintz@rockefeller.edu (NH)

## Introduction

Methyl-CpG-binding-protein 2 (MeCP2) is an abundant nuclear protein that was initially identified as the founding member of the methyl-DNA binding proteins family (*Lewis et al., 1992*). Mutations in the X-linked *Mecp2* gene cause Rett syndrome, a severe neurological disorder characterized by developmental regression during the first few years of life (*Amir et al., 1999*). Although MeCP2 is expressed by all cell types, it is present at very high levels in neurons (*Skene et al., 2010*). Loss of MeCP2 is not lethal to these cells but results in alterations in gene expression and reduced cellular growth (*Armstrong et al., 1995*; *Chahrour et al., 2008*). Although the consequences of MeCP2 loss are well documented, the mechanisms by which MeCP2 functions may include both the recruitment of specific epigenetic regulatory factors to select sites in the genome (*Chahrour et al., 2008*; *Jones et al., 1998*; *Kokura et al., 2001*; *Kruusvee et al., 2017*; *Lyst et al., 2013*) and a more general role as an organizer of chromatin topology in broad nuclear regions (*Baker et al., 2013*; *Della Ragione et al., 2016*; *Linhoff et al., 2015*; *Lyst and Bird, 2015*). Given the abundance of MeCP2 and its important role in the regulation of chromatin structure and gene expression, further investigation of its chromatin interactions in neurons is of interest.

Genome-wide mapping studies of MeCP2 have shown that it is enriched in highly methylated regions, although it can be found also in regions where levels of DNA methylation are low (*Baubec et al., 2013*; *Mellén et al., 2017*; *Skene et al., 2010*). Oddly, despite its high expression in neurons, MeCP2 is not sufficiently abundant to bind simultaneously to all methylated sites in the

neuronal genome. Hence, the mouse genome contains $\sim 2 \times 10^9$ cytosines in the mouse genome (*NHGRI, 2002*) and 5–6% of cytokines are methylated in most cells of the adult body (*Globisch et al., 2010*) including granule cells (*Figure 6—figure supplement 2B*), which means that there are $\sim 10^8$ 5 mC in each nucleus, while there are not more than $2 \times 10^7$ MeCP2 molecules (*Skene et al., 2010*). This suggests that the global distribution of MeCP2 observed in ChIP studies might result from summation of MeCP2-chromatin interactions that are rather sparse with respect to DNA methylation and quite dynamic. This is consistent with previous imaging studies that have employed fluorescence recovery after photobleaching (FRAP) to document the mobility of MeCP2 in the nucleus of culture cell lines (*Agarwal et al., 2011*; *Ghosh et al., 2010a*; *Kumar et al., 2008*). While these indirect measurements of average MeCP2 behavior over large regions of the nucleus indicate that MeCP2 is mobile, they provide little insight into the mechanisms governing MeCP2 target site selection.

Recent single molecule imaging studies of transcription factors in living cells have yielded surprising insights into their dynamic behavior in the nucleus and revealed unanticipated features of their target search strategies that ought to be considered in models of eukaryotic gene regulation (*Chen et al., 2014b*; *Chong et al., 2018*; *Liu and Tjian, 2018*). To gain additional insights into the dynamic behavior of MeCP2 and its interactions with neuronal chromatin, we have applied this approach to the analysis of MeCP2 in the nucleus of living neurons. Measurements of the behavior of single MeCP2 molecules in the nucleus of cerebellar granule cells relative to other well-studied nuclear proteins revealed that the binding and diffusion of MeCP2 in neurons is similar to H1 linker histones, and distinct from both core histones and site-specific transcription factors. Specifically, extensive mutagenesis experiments demonstrate that MeCP2 behavior in the nucleus can be functionally attributed to a stably bound state with short life times (~7 s) and an unbound state with MeCP2 diffusing much slower than site-specific TFs. Interestingly, we found that the fraction of stably bound state is exquisitely sensitive to the level of DNA modification, and is altered by Rett Syndrome mutations that impact the methyl-CpG binding domain (MBD). In contrast, independent interactions governed by the three AT hook domains at the C-terminus of MeCP2 played a dominant role in regulating the diffusion of unbound MeCP2 with relatively minor contributions from MBD. Furthermore, MeCP2 mobility in the nucleus is sensitive to the local chromatin environment in granule cells, and is dramatically accelerated in the large, euchromatic nuclei of Purkinje cells. Together, our results provide a physical model to explain how distinct types of molecular interactions regulate dynamic MeCP2 binding and diffusion in the nucleus that likely impact its function.

## Results

### MeCP2 nuclear dynamics at single molecule resolution in cerebellar neurons reveals its unique slow diffusive behavior

To study MeCP2 at single molecule resolution in live neurons, we transfected mouse cerebellar granule cells with a construct expressing MeCP2-HaloTag fusion protein and cultured them for 12–14 days to generate a monolayer of neuronal cells that is ideal for image acquisition (*Figure 1A*). To measure the behavior of MeCP2 relative to other well studied nuclear proteins, we used in parallel vectors expressing the HaloTag domain fused to the histone H2b, to a small domain containing MeCP2 minimal nuclear localization signal (NLS), to the transcription factors Sox2 and TBP, and to the linker histone H1.0. The cells were then incubated with the fluorescent HaloLigand JF549 to specifically label HaloTagged proteins (*Figure 1B*).

Two different sets of data were collected to describe the behavior of MeCP2 relative to histones and conventional transcription factors. To determine the residence time of the stably bound proteins, we acquired images from sparsely labeled nuclei at low excitation power and high acquisition time (500 ms/frame). As a result, while mobile particles are blurred into the background, immobile single molecules are selectively recorded as individual bright spots that can be followed for several frames until they dissociate from their binding sites (*Figure 1—figure supplement 1*). In agreement with previous studies of nuclear factors interacting with chromatin (*Chen et al., 2014b*), the dissociation kinetics of MeCP2 is best described by a double exponential decay that accounts for short-lived and long-lived populations of bound molecules (*Figure 1—figure supplement 1B*). This dwelling time analysis demonstrates that approximately ~25% of immobile MeCP2 molecules are long-lived,

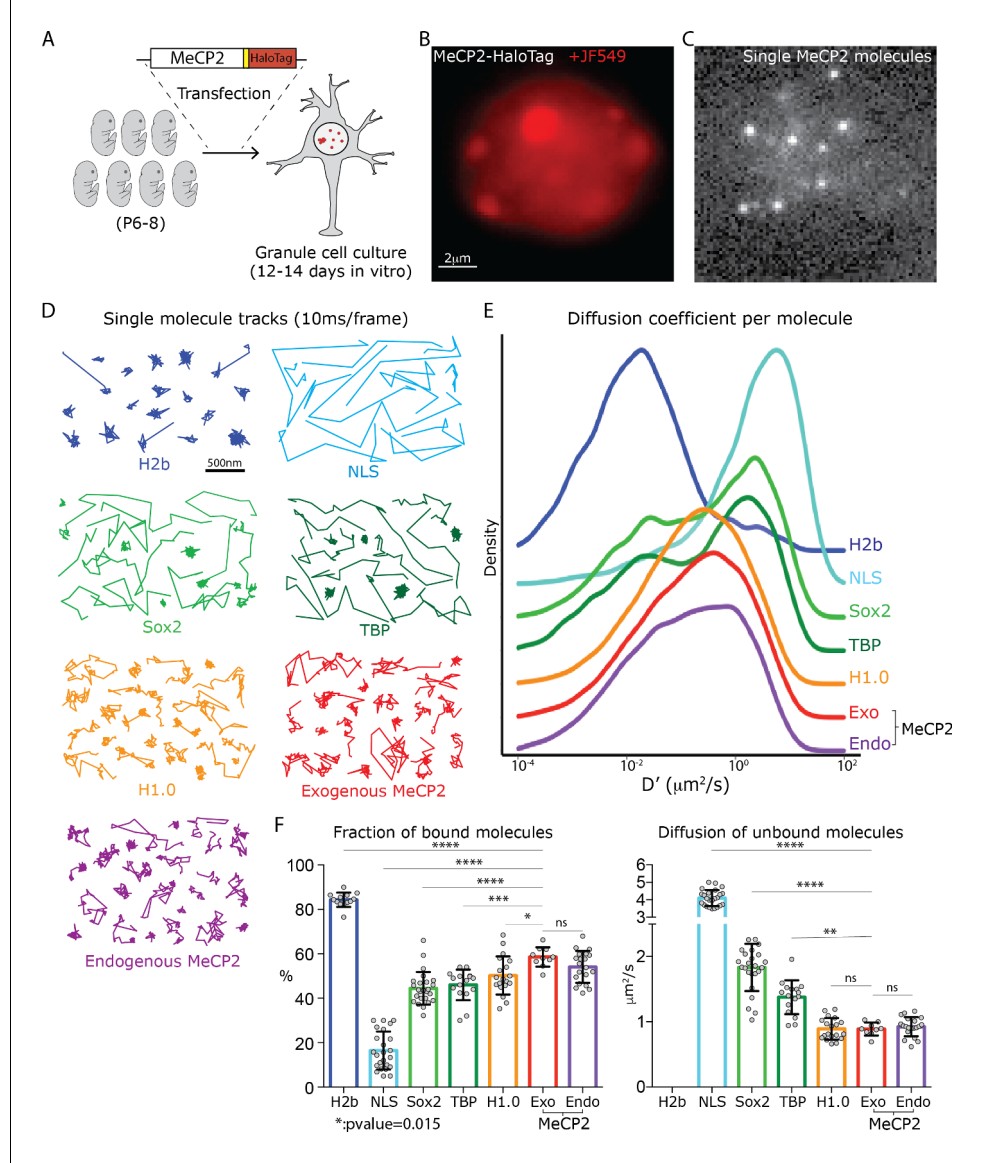

**Figure 1.** Analysis of single molecule kinetics reveals unique the slow diffusive behavior of MeCP2. (**A**) Schematic representation of the experimental strategy used. Cerebellar granule cells (GC) from 6 to 8 day old pups were transfected with constructs expressing the HaloTag domain in frame to MeCP2 coding sequence, or other control proteins. For MeCP2 endogenous expression, 6–8 day old MeCP2-HaloTag knock-in pups were used for granule cell culture under the same experimental conditions. Single MeCP2 molecules were imaged in live GC cultures by (**B**) specifically labeling MeCP2-HaloTag proteins with JF549-HaloLigand, (**C**) photo-bleaching at high laser power to gain sparse labeling and single molecule images were recorded at 100 Hz. (**D**) Representative single molecule tracks obtained from live granule cells nuclei for the different proteins at 100 Hz. (**E**) Distribution of diffusion coefficients (D', $\mu m^2$/s) calculated for individual molecules assuming Brownian motion. H2b (~20,000 molecules, blue), NLS (~10,000 molecules, tourquise), Sox2 (~15,000 molecules, green), TBP (~8000 molecules, dark green), H1.0 (~50,000 molecules, orange), Exogenous MeCP2 (~20,000 molecules, red), and Endogenous MeCP2 (~28,000 molecules, orange). The curves were plot stacked on a joy plot and the origins of the y axis for each curve are displaced for clarity (**F**) Fraction (Left) of bound molecules and average diffusion coefficient (Right) of the unbound molecules calculated for individual cell by fitting the SpotOn 2-state model to the distribution of translocations for individual molecules of H2b (9 cells, blue), NLS (19 cells, turquoise), Sox2 (25 cells, green), TBP (16 cells, dark green), H1.0 (19 cells, orange), Exogenous MeCP2 (20 cells, red) and Endogenous MeCP2 (23 cells, orange). Graphs show mean and standard deviation of single cell value (gray dots) and compare Exogenous MeCP2 to each of other nuclear factors by one-way ANOVA with Bonferroni's multiple comparison test: **** ($p < 0.0001$), * ($p < 0.05, > 0.01$).

*Figure 1 continued on next page*

*Figure 1 continued*

The online version of this article includes the following figure supplement(s) for figure 1:

**Figure supplement 1.** Dwell time analysis of stable single molecules MeCP2 in living neurons reveals highly dynamic interactions with chromatin.

**Figure supplement 2.** Generation of MeCP2-HaloTag knock-in animals.

**Figure supplement 3.** MeCP2 overexpression does not affect its nuclear kinetics.

with residence times of ~7 s. This is shorter than the residence times of Sox2 or TBP (~10 and ~14 s, respectively) or the linker histone H1.0 (~10 s). The majority of MeCP2 binding interactions are shorter lived (~2 s), as is the case with the other factors analyzed. These data demonstrate that MeCP2 engages in two different stable binding interactions with its cognate DNA sites.

To analyze the dynamic behavior of transiently bound or rapidly diffusing MeCP2 molecules, images from sparsely labeled living granule cells (*Figure 1C*) were acquired at very high speed (100 Hz) to generate 2D single molecule tracks of the movements of single tagged proteins in the nucleus. It is evident from these single molecule trajectories (*Figure 1D*) that the dynamics of MeCP2 in granule cell nuclei is quite distinct from stably bound histones (H2b) or sequence specific transcription factors (Sox2 and TBP). Rather, even at this superficial level of inspection, the behavior of MeCP2 and the linker histone H1.0 appear similar. This impression is confirmed by calculation of the distribution of diffusion coefficients calculated for each of these nuclear proteins, revealing their similarity to one another relative to Sox2, TBP, histone H2B, or the NLS control (*Figure 1E*).

To better evaluate the distinctive diffusive properties of MeCP2 molecules, we used the algorithm package Spot-On (*Hansen et al., 2018*). At this high rate of image acquisition, the long and short-lived stably bound proteins revealed in the residence time measurements cannot be distinguished from one another, and both are included in the bound fraction because their diffusion constants are very slow.

Spot-On derives the fraction of stably bound molecules, those that diffuse <0.08 $\mu m^2$/s, and also calculates the diffusion coefficient of those that are unbound and rapidly moving (*Figure 1F*). As expected given their stable incorporation into nucleosomes, the vast majority of H2b molecules (>80%) are immobile, while most of the NLS molecules (80–90%) are very rapidly diffusing (~4 $\mu m^2$/s). As previously shown, a minority of Sox2 and TBP molecules are stably bound. Diffusion of unbound Sox2 and TBP proteins is relatively rapid and similar to that previously reported in ES cells, averaging ~2 and 1.5 $\mu m^2$/s per cell respectively, (*Figure 1F*) (*Chen et al., 2014b*). These data confirm the general rule for sequence-specific transcription factors that less than half of the population is stably bound to DNA, and the rest moves rapidly 'searching' chromatin to find cognate binding sites. Two features distinguish the dynamic behavior of MeCP2 from transcription factors (*Figure 1F*). First, the fraction of MeCP2 molecules bound in a stable mode is significantly higher than that of Sox2 or TBP. Thus, approximately half of the MeCP2 population is stably bound in granule cell nuclei at any given time. Second, when not bound stably, MeCP2 diffuses significantly more slowly (averaging ~1 $\mu m^2$/s per cell) than Sox2, TBP or the NLS proteins. Interestingly, the dynamic behavior of MeCP2 is strikingly similar to the linker histone H1.0, where ~ 50% of molecules are immobile and unbound molecules diffuse slowly inside the nucleus (~1 $\mu m^2$/s per cell) (*Figure 1F*).

## MeCP2 nuclear dynamics do not depend on its expression level

MeCP2 gene dosage is critical for normal neuronal function (*Collins et al., 2004*; *Ramocki et al., 2009*). To establish that the slow diffusive properties we have measured reflect the properties of endogenous MeCP2, we generated a mouse line genetically engineered to harbor an in-frame sequence encoding the HaloTag domain in the 3' UTR of *Mecp2* gene (*Figure 1—figure supplement 2A and B*). Animals expressing the MeCP2-HaloTag fusion protein are viable and show no disease phenotype, demonstrating that the HaloTag domain does not disrupt MeCP2 physiological function. Single molecule tracks of endogenous MeCP2 imaged in granule cells cultured from the knock-in mice resemble those generated by exogenously expressed proteins (*Figure 1D*). Furthermore, the distributions of diffusion coefficients for individual molecules of exogenous and endogenous MeCP2 strongly overlap (*Figure 1E*). Both the bound fraction and the rate of diffusion of the unbound molecules calculated for single cells do not differ significantly between the two conditions

(*Figure 1F*), indicating that endogenous and exogenous MeCP2 have identical nuclear dynamics in granule cells.

As an additional control to directly demonstrate identical dynamic properties of endogenous and exogenously introduced MeCP2, we co-cultured MeCP2-HaloTag granule cells transfected with a vector expressing a MeCP2-GFP fusion protein with untransfected MeCP2-HaloTag granule cells (*Figure 1—figure supplement 3A*). In this way, MeCP2 molecules could be acquired from cells expressing them at endogenous level (GFP negative) and, in parallel from cells containing additional copies of exogenously expressed MeCP2 proteins (GFP positive, *Figure 1—figure supplement 3B*). No difference between these two conditions could be detected in the distribution of MeCP2 diffusion coefficients (*Figure 1—figure supplement 3C*), the fraction of bound molecules or diffusion rate of unbound molecules per cell (*Figure 1—figure supplement 3D*). Furthermore, the dwell time analysis revealed comparable behavior for MeCP2 regardless of the expression system (*Figure 1—figure supplement 3E*).

Taken together, our data demonstrate that the dynamic behavior of MeCP2 in granule cell nuclei is distinct from both stably bound histones and typical sequence-specific transcription factors. It is also established that both the endogenously and exogenously expressed MECP2-HaloTag proteins are suitable for in depth investigation of both its bound state, and the factors contributing to the peculiar, slow diffusion of unbound MeCP2 in neuronal nuclei.

## MeCP2 chromatin binding is sensitive to DNA modification levels in neurons

Biochemical studies initially determined that the ability of MeCP2 to bind DNA depends on the presence of methylated cytosines (5mC) (*Lewis et al., 1992*) and more recently, it was shown that hydroxylation of these sites (5hmC), when it occurs in the context of CpG, disrupts this interaction (*Figure 2—figure supplement 1A*) (*Ayata, 2013*; *Gabel et al., 2015*; *Mellén et al., 2017*). Once neurons become postmitotic, they begin accumulating de novo DNA methylation sites by the activity of DNA methyltransferase 3a (DNMT3a), as well as converting 5mC to 5-hyroxymethylcytosine (5hmC) through the enzymes Tet1, 2 and 3 (*Lister et al., 2013*; *Stroud et al., 2017*). To understand the relationship between these DNA modifications and MeCP2 dynamics in live granule cells, we removed these epigenetic factors using appropriate conditional alleles and Cre drivers, and we observed MeCP2 behavior by expressing Double-floxed Inverted Orientation (DIO) MeCP2-HaloTag (*Figure 2A*).

We employed the Neurod1 Cre driver line in Dnmt3a[flox/flox] animals to generate granule cells lacking de novo DNA methylation (Neurod1Cre;Dnmt3a[flox/flox], Dnmt3a_cKO, *Figure 2A*), which resulted in ~20% decrease of total 5mC level in both CG and non-CG sequence contexts (*Figure 2—figure supplement 1B,C,D and E*). MeCP2 nuclear trajectories recorded in these hypo-methylated granule cells display increased mobilty as shown by cumulative distribution of D' values for individual molecules (*Figure 2—figure supplement 1G*). The fraction of bound MeCP2 in the Dnmt3a KO cells decreased from ~55% to~45%, in agreement with the ~20% depletion of DNA methylation in these cells (decrease from ~6% to 5% of all cytosines, *Figure 2—figure supplement 1C*). In contrast, the rate of diffusion of unbound MeCP2 molecules is largely unaffected by this decrease in DNA methylation (*Figure 2B*).

To prevent DNA oxidation of 5mC to 5hmC in granule neurons, we depleted the three Tet oxidases by transfecting a Cre expressing vector in cultured granule cells from Tet1[flox/flox];Tet2[flox/flox];Tet3[flox/flox] animals (*Li et al., 2016*) (*Figure 2A*). Because a DIO-MeCP2-HaloTag construct was co-transfected into these cells, MeCP2-HaloTag could only be observed in cells with impaired Tet activity (Tets_cKO) and in turn devoid of 5hmC sites (*Figure 2—figure supplement 1F*). Loss of expression of Tet proteins results in a subtle reduction in MeCP2 nuclear mobility as observed by the distribution of diffusion coefficients for individual molecules (*Figure 2—figure supplement 1G*). While the diffusion of unbound MeCP2 molecules remains essentially unchanged in cells lacking 5hmC, the fraction of bound molecules increases from ~55% to~60%, (*Figure 2B*). This is consistent with the higher number of high affinity 5mCG binding sites expected as a result of the failure to convert them to low affinity 5hmCG sites. We note that the small magnitude of this change is not surprising since less than 0.5% of all cytosines are hydroxylated in the granule cell genome (*Figure 2—figure supplement 1C*).

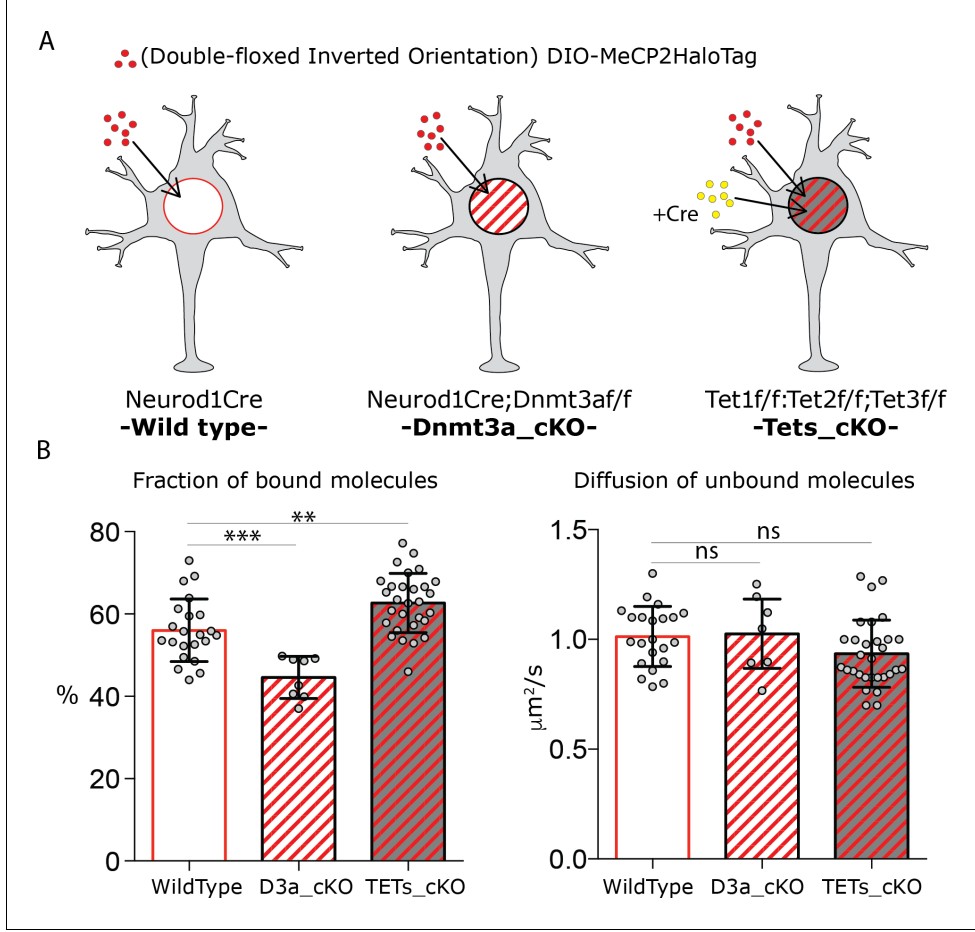

**Figure 2.** DNA modifications regulate MeCP2 binding without affecting its nuclear diffusion. (**A**) Schematic representation of the experimental design. Cre-dependent expression vector carrying DIO-MeCP2-HaloTag was transfected in GC cultures from Neurod1-cre (wild type) or Neurod1-cre;Dnmt3a$^{flox/flox}$ (Dnmt3a_cKO) animals. Cre-expressing vector was co-transfected with DIO-MeCP2-HaloTag construct in GC culture from Tet1$^{flox/flox}$; Tet2$^{flox/flox}$; Tet3$^{flox/flox}$ animals (Tets_cKO (**B**) Fraction of bound molecules and diffusion coefficient of the unbound molecules calculated for individual cells by SpotOn in wild type (22 cells), Dnmt3a depleted (7 cells) and Tet1, 2 and 3 triple knock-out (29 cells) GCs. Graphs show mean and standard deviation of single cell values (gray dots) and compare wild type neurons to mutants by one-way ANOVA with Bonferroni's multiple comparison test: ** (p<0.005); *** (p<0.0005).

The online version of this article includes the following figure supplement(s) for figure 2:

**Figure supplement 1.** Cerebellar Granule cells lacking Dnmt3a or Tet1, Tet2 and Tet3 expression result in altered DNA methylations levels.

Taken together, these data suggest that MeCP2 behavior in live neurons is tightly regulated by epigenetic modifications of the DNA. Thus, as predicted from previous biochemical studies (*Lewis et al., 1992*; *Mellén et al., 2017*), a decrease in the level of DNA methylation as a result of Dnmt3a deletion causes a reduction in the fraction of stably bound MeCP2 in vivo. Conversely an increase in DNA methylation as a consequence of deletion of the Tet oxidases results in an increase in the fraction of stably bound MeCP2. The slow rate of diffusion of unbound MeCP2 in vivo is not dependent on DNA methylation and, therefore, must reflect distinct biochemical properties of MeCP2.

# MeCP2 MBD is required for stable binding and constrains its diffusion in granule cell nuclei

To investigate further the contribution of DNA binding to MeCP2 dynamic behavior, we targeted the MBD domain with two common Rett syndrome mutations, known to disrupt MeCP2 DNA binding to different degrees (*Cuddapah et al., 2014*): R106W, which completely abolishes DNA binding and R133C which retains significant DNA binding capacity (*Nikitina et al., 2007*). R106W-HaloTag DIO and R133C-HaloTag DIO were expressed in cultured NeuroD1Cre+ granule cells in parallel with wild type MeCP2 (*Figure 3A*). As expected, given their distinct DNA binding properties, the two MeCP2 mutants display different behaviors in the nucleus of cultured granule cells. The nuclear distribution of R106W appears homogenous in the nucleus of granule cells, and its nuclear trajectories differ dramatically from wild type. In contrast, the distribution of R133C in the nucleus and its single molecule tracks resemble wild type MeCP2 (*Figure 3B*).

Quantitative analyses of the behavior of single molecules for each of these forms of MeCP2 confirm these observations (*Figure 3B,C,D*). The distribution of diffusion coefficients calculated for individual molecules shows that both mutations result in increased MeCP2 mobility, but to different extents. In particular, SpotOn analysis of R106W molecules demonstrates that the loss of DNA

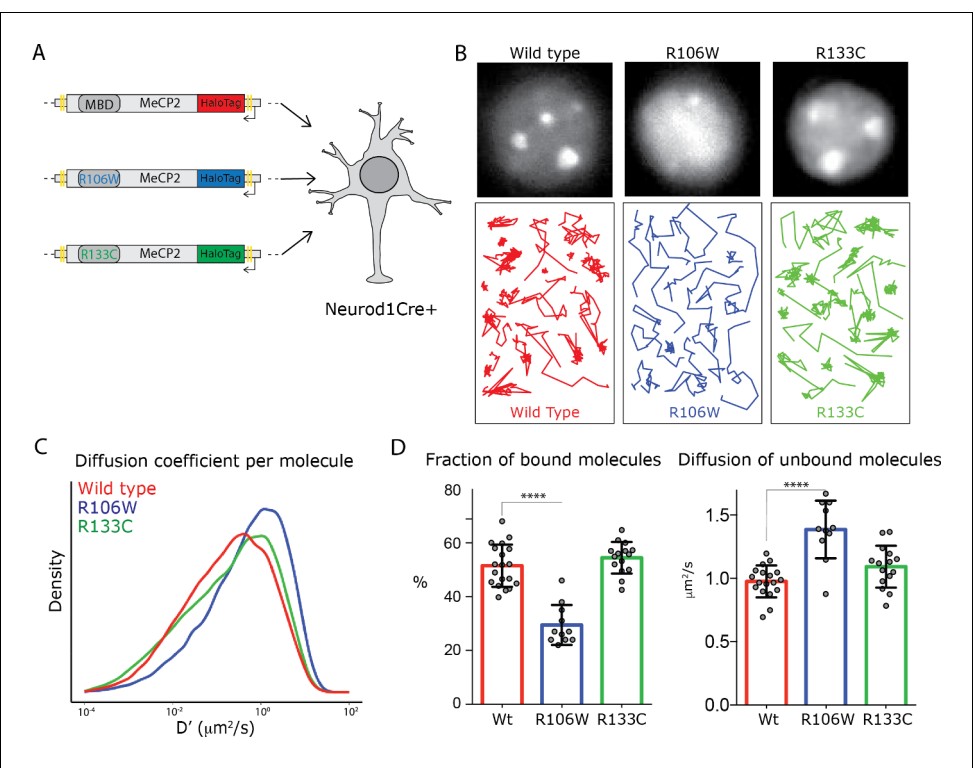

**Figure 3.** Rett-syndrome mutations within the MBD domain disrupt MeCP2 nuclear kinetics. (**A**) Schematic representation of the experimental design. GC culture of Neurod1-cre animals were transfected with DIO constructs carrying wild type MeCP2-HaloTag (red) or bearing the Rett-syndrome mutations Arginine 133 to Cysteine (R133C, green) or Arginine 106 to Tryptophan (R106W, blue). (**B**) Cells labeled with JF549- HaloLigand show the nuclear distribution of wild type MeCP2HaloTag and its mutants (top panels). Representative single molecule tracks of wild type (red) R133C (green) and R106W (blue) in wild type granule cell nuclei. (**C**) Distribution of diffusion coefficients (D', $\mu m^2/s$) calculated for individual molecules of wild type MeCP2 (~18,000 molecules, red), R133C (~16,000 molecules, green) and R106W (~20,000 molecules, blue). Two-sample Kolmogorov-Smirnov test was used to compare wild type MeCP2 with R106W (p-value<$9.7e^{-70}$) and R133C molecules (p-value=$9.7e^{-29}$). (**D**) Fraction of bound molecules and average diffusion coefficient of the unbound molecules calculated in individual cells for wild type MeCP2 (5 cells, red), R1033C (19 cells, green) and R106W (20 cells, blue). Graphs show mean and standard deviation of single cell values (gray dots) and compare wild type MeCP2 to Rett-syndrome mutants by one-way ANOVA with Bonferroni's multiple comparison test: * (p<0.05), **** (p<0.0001).

binding capacity in this mutant strongly decreases the fraction of stably bound protein and results in a significant increase in its rate of diffusion in the nucleus (from ~1 $\mu m^2$/s to ~1.4 $\mu m^2$/s).

The behavior of the R133C mutation is quite different. As anticipated from its nuclear distribution and the appearance of its single molecule nuclear trajectories, the fraction of stably bound R133C does not differ significantly from wild type MeCP2. Interestingly, however, subtle differences in the rate of diffusion of the unbound R133C protein can be observed. This is apparent from both the shift in distribution of the diffusion constants presented in *Figure 3C*, and the trend toward an increased average diffusion coefficient for the entire population from ~1 μm2/s to ~1.2 μm2/s (*Figure 3D*). These experiments confirm the expectation that stable binding of MeCP2 in living nuclei requires the MBD, and they suggest that the MBD may also participate in interactions that constrain its diffusion when it is not stably bound to DNA, perhaps as it samples potential sites for stable binding.

## MeCP2 AT-Hook domains constrain diffusion without affecting DNA binding

The finding that diffusion of unbound MeCP2 resembles the slow diffusion of histone H1.0 rather than the rapid diffusion typical of unbound conventional transcription factors prompted us to investigate which domains of MeCP2 contribute to this interesting behavior. The MBD is encoded by less than 25% of the MeCP2 sequence and structural studies have revealed that most of the protein is composed of Intrinsically Disordered Domains (IDDs) (*Figure 4A*). Rett syndrome nonsense mutations involve many domains outside of the MBD (*Figure 4A*) (*Krishnaraj et al., 2017*), in agreement with previous studies establishing the importance of several of these in MeCP2 interaction with chromatin and function (*Ghosh et al., 2010b*; *Lyst and Bird, 2015*). In addition to three small AT-Hook domains, MeCP2 IDDs comprise the N-terminal sequence (N-term), an Intervening domain (ID) which connects MBD to the Transcriptional Repressive Domain (TRD), containing the nuclear localization signal (NLS) and the NCoR interaction domain (NID), and a C-terminal domain (C-term). To examine the impact of these domains on MeCP2 dynamics in living neuronal nuclei, we transfected Neurod1-Cre+ cerebellar granule cells with DIO constructs expressing the HaloTag in frame with several IDD depleted versions of MeCP2 protein in parallel with full-length wild type MeCP2 as a control (*Figure 4B*). Cultured granule cells expressing these constructs were imaged in the same experimental conditions described above to generate single molecule trajectories, which were then analyzed to measure the fraction of stably bound molecules and the diffusion rate of unbound molecules (*Figure 4C*).

Minimal MeCP2 proteins containing only the MBD and NLS maintains the level of DNA binding relative to wild type control, but when unbound they diffuse significantly faster (~1.5 μm2/s) than full-length MeCP2 (~1 μm2/s). This confirms that the MBD is sufficient for stable DNA binding in vivo, and indicates that additional IDDs are involved in weak interactions that limit MeCP2 nuclear diffusion but do not contribute significantly to stable binding. Introduction of the inactivating R106W mutation into the MBD of this minimal construct reduces stable binding to background levels, and increases further the diffusion of unbound molecules (~2 μm2/s). When taken together with the effects of the R106W and R133C MBD mutations on full length MeCP2, these data demonstrate that the MBD is necessary and sufficient for stable DNA binding in granule cells, that it partially constrains diffusion of unbound MeCP2, and that additional domains of the protein must also contribute to the diffusive properties of MeCP2.

To identify additional domains of MeCP2 that determine its slow diffusion in living granule cell nuclei, we progressively added individual IDDs back to the minimal R106W protein and measured their effects on MeCP2 nuclear diffusion. The addition of the N-terminal domain (+N-term) does not change the dynamic behavior of the protein. Inclusion of the ID domain (+ID), which contains an AT-hook domain, significantly reduces diffusion of unbound molecules. The incorporation of a five amino acid sequence (RGRKP) at residue 268 reconstitutes a second AT-hook domain (+AT-Hook2) but does not affect MeCP2 diffusion rate. However, if AT-hook2 is assayed independently by in-frame fusion to the rapidly diffusing NLS-HaloTag molecules, it is capable of slowing diffusion in granule cell nuclei. (*Figure 4—figure supplement 1A,B and C*). The inclusion of the NID domain further constrains MeCP2 nuclear mobility to a diffusion rate comparable to full length R106W proteins (~1.4 $\mu m^2$/s). Restoration of the full-length protein with the addition C-terminal domain, including a third AT-hook, does not affect MeCP2 diffusion (*Figure 4C*). Finally, since there is a great deal of evidence that MeCP2 AT-hook domains play a significant role in its function (*Baker et al., 2013*;

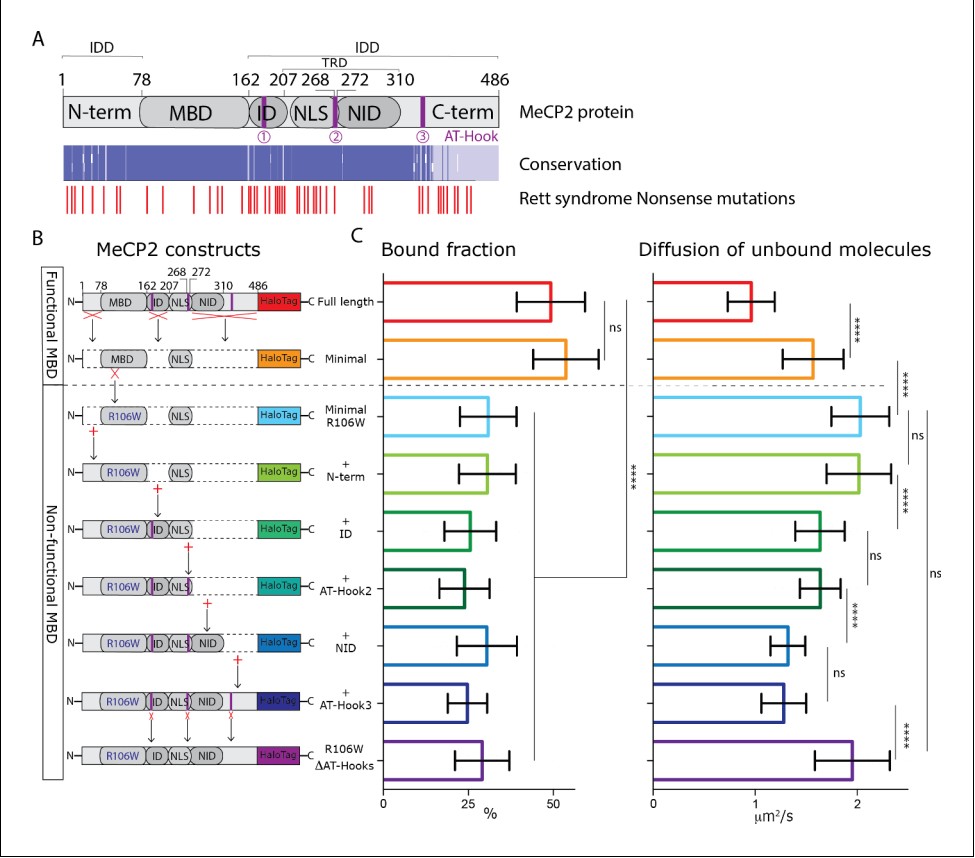

**Figure 4.** MeCP2 AT-Hook Domains hinder its nuclear diffusion without affecting its binding. (**A**) (Top) Schematic representation of full-length MeCP2 protein where known domains are annotated and the 3 AT-Hook domains are highlighted in purple. (Middle) Heat map representing protein sequence identity (from blue, high identity score, to white, no identity) between Human, Mouse, Rat and Dog MeCP2. (Bottom) Missense and nonsense Rett-syndrome causing mutation from RettBASE are annotated in red. (**B**) Diagrams of different DIO constructs used in the experiment, where MeCP2 was sequentially mutated. (**C**) Fraction of bound molecules and average diffusion coefficient of the unbound molecules calculated in individual cells by SpotOn for the different MeCP2 constructs. Full-length (red, 29 cells), Minimal (orange, 31 cells), Minimal R106W (cyan, 25 cells), +N-term (light green, 17 cells), +ID (green, 32 cells), + AT-Hook 2 (dark green, 27 cells), +NID (blue, 26 cells), +AT-Hook3 (dark blue, 11 cells), R106W-ΔAT-Hooks (purple, 32 cells). Graphs show mean and standard deviation of single cell values and compare the different constructs by two-tailed unpaired t-test: **** (p<0.0001).

The online version of this article includes the following figure supplement(s) for figure 4:

**Figure supplement 1.** MeCP2 AT-Hook two domain hinders free diffusion of NLS molecules in the nucleus.

---

*Sheikh et al., 2018*), we tested whether these alone were sufficient to determine the slow diffusion rate of unbound MeCP2. Surprisingly, the full length R106W protein lacking all three AT-hooks displays identical dynamic behavior to the minimal R106W protein that solely contains a non-functional MBD (*Figure 4C and D*). Taken together, these results demonstrate that the IDDs of MeCP2 are not required for stable DNA binding in the nucleus of granule cell neurons. Rather, we find that they engage in weak, transient interactions with genomic DNA or chromatin. In particular, our data highlight the MeCP2 AT-hook domains as important players in the constraint of MeCP2 diffusion in the nucleus.

## MeCP2 nuclear dynamics are sensitive to the level of chromatin compaction

The structure function studies we have presented demonstrate that the dynamic behavior of MeCP2 in the nuclei of living granule cells resembles the linker histone H1.0, that it is sensitive to DNA modifications, and that it depends on both its MBD and the IDDs. The MBD is sufficient for stable

binding of MeCP2 in the nucleus, and independently contributes to its slow diffusion when unbound. By contrast, the AT-hook domains of MeCP2 provide major constraints to the mobility of unbound MeCP2 in granule cells, without generating significant stable binding. Since these domains are known to mediate non-specific interactions that occur in the minor groove of AT-rich DNA independently from DNA methylation, and MBD engages in specific binding to methylated cytosines, the behavior of MeCP2 is most likely the result of a combination of specific and non-specific interactions. Given the increased DNA concentration and the enrichment of methylated DNA in highly compact heterochromatin, we next tested the diffusive properties of MeCP2 in euchromatic and heterochromatic domains of the granule cell nucleus.

To identify compact chromatin domains in living nuclei, we transfected a fusion of heterochromatin protein one alpha (HP1α) in frame with the Blue Fluorescent Protein (HP1α-BFP) in MeCP2-HaloTag granule cells. In this way, it is possible to identify regions in the nucleus of living neurons that are highly compact (heterochromatic, HP1α positive) or more loosely arranged (euchromatic, HP1α negative) (*Figure 5A*). From snapshot acquisitions of HP1α-BFP signal we generated filtering masks, which allowed us to discriminate single molecule tracks recorded within heterochromatin (inside the mask, red), and those occurring within euchromatin (outside the mask, green) (*Figure 5B*). Although the differences in the two nuclear sub-regions are not large enough to be assessed at the population level (*Figure 5E*), the distributions of the diffusion coefficients of individual molecules show that euchromatic MeCP2 molecule are significantly more dynamic than those present in heterochromatin (*Figure 5C*) even when the two regions a compared within the same cell (*Figure 5D*). Furthermore, dwell time analysis shows that while MeCP2 residence times remain unchanged, the fraction of long-lived bound molecules is significantly higher in heterochromatin than euchromatin, consistent with the elevated number of MeCP2 binding sites (*Figure 5F*).

## MeCP2 mobility depends on the cell type and nuclear architecture

Despite the very compact nature of granule cell nuclei, we observed subtle differences in MeCP2 dynamic behavior that correlate with the level of chromatin compaction. To further probe this observation, and to determine whether the properties of MeCP2 change in cell types whose nuclear architecture varies significantly from granule cells, we chose to compare its behavior in granule cells and Purkinje cells (*Figure 6A*, *Figure 1—figure supplement 2*). We prepared acute slices from the cerebellum of the MeCP2-HaloTag animals, incubated them with the JF549 ligand, and imaged MeCP2 single molecule trajectories in granule cells and Purkinje cells using lattice light sheet microscopy (*Chen et al., 2014a*).

The dynamic features of MeCP2 in granule cells from acute cerebellar slices are remarkably similar to culture experiments. Although the fraction of stably bound MeCP2 is slightly higher in the acute slices prepared from adult mice compared to cultured granule cells, the slow diffusion of unbound MeCP2 is indistinguishable in the two conditions (*Figure 6*, *Figure 6—figure supplement 1*). The dynamic behavior of MeCP2 in the large, euchromatic nuclei of Purkinje cells is very different. As is evident from the distinct appearance of single molecule tracks in Purkinje cells (*Figure 6A*), the cumulative distribution of the diffusion coefficients for individual molecules in Purkinje cells demonstrates that MeCP2 is significantly more mobile in the large, euchromatic Purkinje cell nuclei (*Figure 6B*). Furthermore, while the fraction of stably bound MeCP2 molecules is similar between these cell types (70 vs. 60%), unbound MeCP2 diffuses considerably faster (~2X) in PC than it does in GC (*Figure 6C*). Finally, the dwell time analysis shows a higher fraction of long-lived immobile molecules in GC compared to PC (*Figure 6—figure supplement 1B*). These data demonstrate that nuclear architecture and chromatin compaction play major roles in the dynamic behavior of MeCP2, and suggest that differences in these properties between cell types may influence MeCP2 function in as yet unexplored ways.

## Discussion

Recent single molecule imaging studies have provided important insights into transcription factor activity that complement those obtained from biochemical, genomic and ensemble imaging studies (*Chen et al., 2014b*; *Liu and Tjian, 2018*). Here we have adopted this approach to measure the dynamic behavior of the fundamentally important neuronal methyl-CpG binding protein (MeCP2) in live neurons. The data we have collected both confirm previous studies of the structure and function

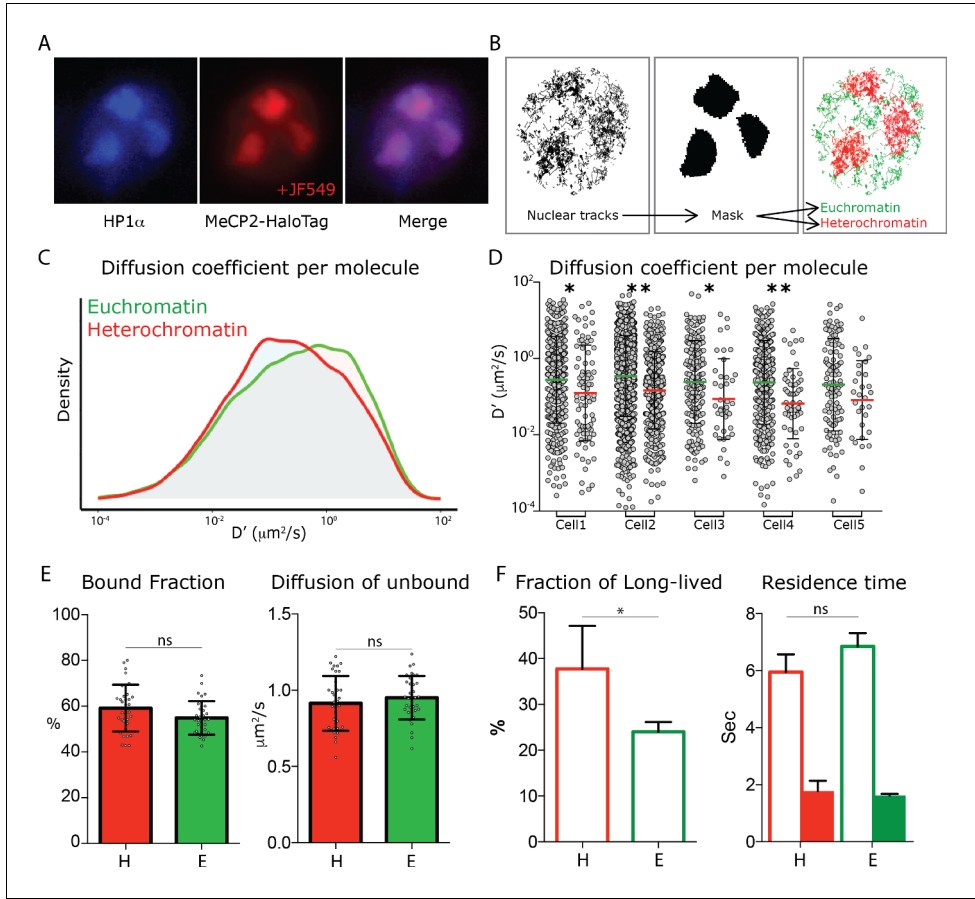

**Figure 5.** MeCP2 diffusion in euchromatin and heterochromatin regions. (A) GC culture of MeCP2HaloTag animals was transfected with a construct expressing HP1αBFP (blue) and labeled with JF549-HaloLigand (red). (B) Nuclear single molecule tracks are spatially filtered through masks generated from whole MeCP2HaloTag-JF549 fluorescence signal to separate them into euchromatin (green) and heterochromatin (red). (C) Distribution of diffusion coefficients (D', μm²/s) calculated for individual MeCP2 molecules in euchromatin (~22,000 molecules, green) or in heterochromatin (9000 molecules, red) of live granule cell nuclei. Two-sample Kolmogorov-Smirnov test was used to compare MeCP2 molecules in euchromatin and heterochromatin p-value=4.6e⁻²² (D) Scatter dot plot of Diffusion coefficients (D', μm²/s) for inidvidual molecules in euchromatin (green) and heterochromatin (red) within the same cells. Two-sample Kolmogorov-Smirnov test was used to compare MeCP2 molecules in euchromatin and heterochromatin: * (p<0.05), ** (p<0.005). (E) Bound fraction and average diffusion coefficients of the unbound molecules calculated in each cell by SpotOn for MeCP2 in euchromatin (E, green) and heterochromatin (H, red) for 15 cells. Graphs show mean and standard deviation of single cell values (gray dots) and compare MeCP2 euchromatin and heterochromatin regions by two-tailed unpaired t-test: ns (p>0.05). (F) Dwell time analysis of stable MeCP2 particles as described in *Figure 1—figure supplement 2* in euchromatic and heterochromatic nuclear sub-regions. Graph show Long-lived fraction (left) and resident time of the Long-lived molecules (right) of stable MeCP2 for three independent replicates. Graphs show mean and standard deviation and compare Heterochromatin and Euchromatin by two-tailed unpaired t-test: * (p<0.05).

of MeCP2, and reveal interesting new features that are important to consider in models of its roles in neuronal nuclei.

## The dynamic behavior of MeCP2 in granule cells resembles histone H1.0 but is highly sensitive to the level of DNA methylation

Previous studies using FRAP have reported that in cultured cell lines the overall mobility of MeCP2 and histone H1.0 are similar (*Ghosh et al., 2010a*). Our measurements from single molecule imaging experiments of both endogenous and exogenously expressed MeCP2 in primary cultures of cerebellar granule neurons, as well as in acute slice preparations from adult mouse cerebellum, confirm and

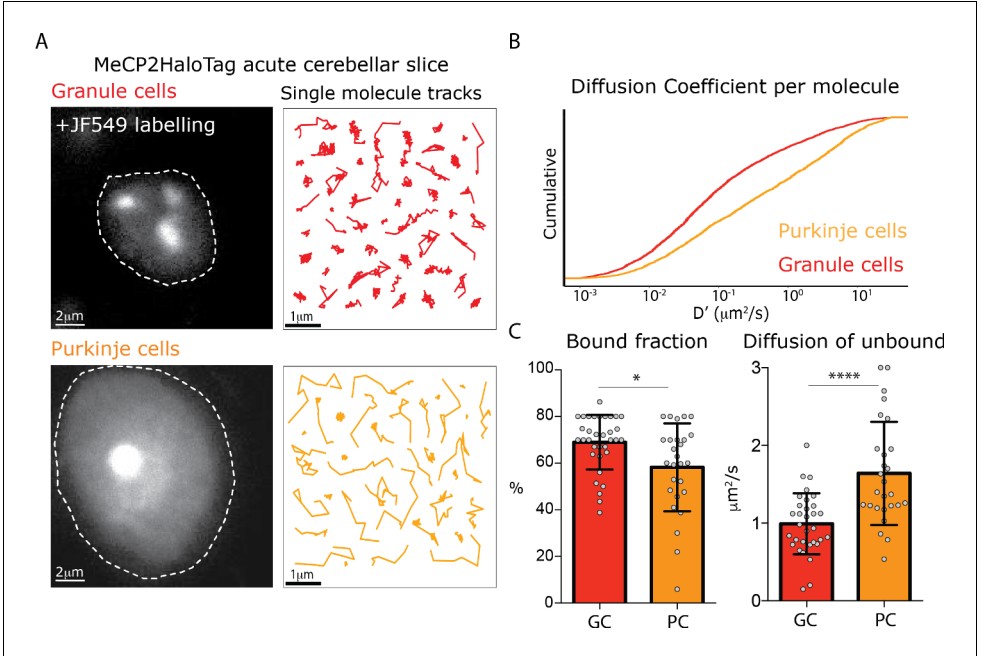

**Figure 6.** Different MeCP2 dynamic behavior in Granule and Purkinje cells from acute brain slices. (**A**) Representative Granule and Purkinje cell nuclei in acute cerebellar brain slices from MeCP2-HaloTag adult mice labeled with JF549-HaloLigand. (**B**) Cumulative distribution of diffusion coefficients in the logarithmic scale (D', μm$^2$/s) calculated for individual MeCP2 molecules in Granule (red) and Purkinje nuclei (orange) from acute cerebellar brain slice. Two-sample Kolmogorov-Smirnov test was used to compare the Diffusion Coefficient calculated for individual MeCP2 molecules in Purkinje and Granule cells. p-value 4.79e$^{-68}$ (**C**) Fraction of bound molecules and average diffusion coefficient of the unbound molecules of MeCP2 determined in individual granule cells (GC) and Purkinje cells (PC). Graphs show mean and standard deviation of single cell values (gray dots) comparing the two cell types by two-tailed unpaired t-test: * (p<0.05), **** (p<0.0001).

The online version of this article includes the following figure supplement(s) for figure 6:

**Figure supplement 1.** Single molecule analysis of MeCP2 in acute brain slices.
**Figure supplement 2.** Different amounts of DNA modification in Granule and Purkinje cells of the mouse cerebellum.

extend these observations. The analysis of individual nuclear trajectories demonstrates that both H1.0 and MeCP2 are less stable than the core histone H2b, but are not as mobile as sequence specific transcription factors Sox2 and TBP. Although similarly to Sox2 and TBP only a fraction of MeCP2 and H1.0 is immobile in neuronal nuclei, the rates of diffusion of the unbound molecules are much slower than these transcription factors, indicating that MeCP2 and H1.0 are somehow more productively and continuously engaging in interactions with chromatin (*Figure 1*). Despite the similar, constrained diffusion of MeCP2 and histone H1.0, a larger fraction of MeCP2 is engaged in long-term stable binding events than H1.0, although its residence time when stably bound is significantly shorter (*Figure 6—figure supplement 1*).

In contrast to linker histones, MeCP2 binding is strongly regulated by DNA methylation. While it can interact non-specifically with DNA with low affinity, MeCP2 high affinity interactions depend on the presence of 5mC sites (*Baubec et al., 2013*; *Meehan et al., 1992*), which are masked by conversion to 5hmC (*Mellén et al., 2017*). ChIP assays from fixed neuronal tissue and native genome wide mapping studies have shown that MeCP2 genome occupancy in neurons is cell type specific and correlated with the status of cytosine modification (*Mellén et al., 2017*). The live cell imaging data we present here confirm these results, and reveal that the behavior of single MeCP2 molecules in neuronal nuclei respond quantitatively to alterations in DNA modification. Thus, without affecting the diffusion of unbound molecules, the balance of Dnmt3a and Tet activities tightly modulates the amount of available high affinity binding sites and dictates the magnitude of stably bound MeCP2 molecules in each cell (*Figure 2*).

The fact that the fraction of MeCP2 bound stably in live neurons is very sensitive to alterations in high affinity 5mC binding sites, despite their large excess (~10X) in the genome relative to the number of MeCP2 molecules, indicates that MeCP2 dynamic behavior is determined by the probability of encountering these targets relative to low affinity sites rather than by the absolute levels of DNA methylation. This suggests that epigenetic modifications accumulating on the DNA of postmitotic neurons determine the local enrichment, and presumably function, of MeCP2 in neuronal nuclei.

## Different aspects of MeCP2 dynamic behavior are governed by distinct protein domains

Several functional domains have been identified previously in MeCP2 and shown to play important roles in vivo and in vitro (*Baker et al., 2013*; *Ghosh et al., 2010b*; *Heckman et al., 2014*; *Tillotson et al., 2017*). Here we have examined the contributions of these domains to the dynamic behavior of single MeCP2 molecules in live granule cell nuclei, revealing several interesting features of MeCP2 that have not been reported previously.

### The MeCP2 Methyl-CpG Binding Domain

Several mutations that cause Rett Syndrome disrupt high affinity binding of MeCP2 to DNA (*Ballestar et al., 2000*; *Yusufzai and Wolffe, 2000*). The R106W mutation, which has been reported previously to eliminate high affinity binding of MBD (*Ballestar et al., 2000*) and to result in a very severe clinical outcome (*Cuddapah et al., 2014*), dramatically decreases the binding of MeCP2 in live neurons whether in the context of the intact protein or a minimal MBD isoform (*Figures 3* and *4*). Interestingly, this single point mutation also increases the diffusion of unbound MeCP2, indicating that the MBD also constrains movement of MeCP2 in the nucleoplasm. Mutation R133C causes a mild form of Rett Syndrome (*Cuddapah et al., 2014*), and has a rather attenuated phenotype when knocked in to the mouse genome (*Brown et al., 2016*; *Mellén et al., 2012*). In our single molecule studies in live neurons, this mutation has no discernable effect on the fraction of MeCP2 bound stably to DNA. Rather, R133C alters the rate of diffusion of MeCP2 when not stably bound in vivo. This agrees with prior studies indicating that R133C does not disrupt high affinity binding to methylated DNA (*Mellén et al., 2012*; *Brown et al., 2016*), and provides additional support for a role of the MBD in constraining the movement of unbound MeCP2 suggesting that alterations in the dynamic behavior of MeCP2 when not stably bound to DNA may also have important functional consequences in its mode of diffusion.

### The MeCP2 Intrinsically Disordered Domains

Numerous studies have shown that MeCP2 requires both an intact MBD and its C-terminal intrinsically disordered domains (IDDs) to execute its function (*Lyst and Bird, 2015*). Here we show that MeCP2 lacking IDDs maintains its DNA binding properties, but diffuses faster, indicating that the MBD is sufficient to bind high affinity target sites while IDDs likely engage in low affinity interactions, possibly with DNA as well as other proteins in the nucleus. To assess which of these additional domains of MeCP2 are important for determining its dynamic properties in granule cell nuclei, we have assayed their independent contributions in the context of the R106W MBD mutation that eliminates stable DNA binding. The inclusions of the Intervening Domain (ID) and the NCoR/SMRT Interaction (NID) are sufficient to restore the slow rate of diffusion of the full-length protein (*Figure 4*), corroborating previous studies showing that ID binds non-specifically to DNA at low affinity (*Claveria-Gimeno et al., 2017*; *Ghosh et al., 2010b*) and that the NID domain is an important hub for the recruitment of epigenetic factors (*Lyst et al., 2013*) or directly interacts with AT-rich DNA trough its basic cluster (*Heckman et al., 2014*; *Mushtaq et al., 2018*). This is also in agreement with previous reports showing by FRAP that ID and NID (TRD) contribute to MeCP2 slow dynamic behavior in culture cell lines (*Kumar et al., 2008*).

### The MeCP2 AT-Hook Domains

In the C-terminal portion of its amino acid sequence, MeCP2 contains three small AT-hook DNA binding domains within and surrounding the ID and NID (*Figure 4A*). Earlier investigations of these domains have demonstrated that they are involved with MeCP2 function (*Baker et al., 2013*) and, similar to other AT-hook domains, interact preferentially with AT rich DNA (*Reeves and Nissen,*

*1990*). Here we have shown that these low affinity interactions alone can determine the slow diffusion of MeCP2 in granule cells. Thus, lack of AT-hooks eliminates ID and NID contributions to MeCP2 dynamic behavior, indicating that ID requires AT-hook one to bind DNA and that NID requires at least two AT-hook domains to assemble protein complexes independently from MBD (*Figure 4*). This demonstrates that the AT-hook domains are critically important determinants influencing the dynamic behavior of MeCP2 in living neurons, and they strongly support the hypothesis that they play an important role in MeCP2 function (*Baker et al., 2013*).

## MeCP2 dynamic behavior is sensitive to local chromatin structure

The dynamic behavior of MeCP2 in granule cell nuclei results from both high affinity specific interactions with methylated DNA and non-specific low affinity interactions with chromatin. The probability of encountering these targets in the nucleoplasm determines MeCP2 behavior, and maximizes transient local interactions as MeCP2 samples DNA for its high affinity binding sites. Granule cell nuclei are among the most compact in the nervous system. Active regions of the granule cell genome are euchromatic, and they are interspersed with large regions of highly compact, inactive heterochromatin. Surprisingly, MeCP2 binding and mobility in euchromatin and heterochromatin is very similar. Although the average diffusion coefficient of MeCP2 molecules is lower in heterochromatin than in euchromatin (*Figure 5C and D*), there is little apparent impact on the fraction of stably bound MeCP2 or residence times in these different chromatin domains (*Figure 5E and F*).

In light of the very subtle differences we have observed in MeCP2 dynamics in granule cell heterochromatin and euchromatin, we were surprised at the dramatic differences in MeCP2 dynamic behavior we observed in the large, euchromatic nuclei of Purkinje cells. While the decrease in the fraction of bound MeCP2 in Purkinje cells can be explained by the decrease in methylation and increase in hydroxymethylation of Purkinje cell DNA (*Figure 6—figure supplement 1*), the rapid diffusion of unbound MeCP2 in live Purkinje cell nuclei resembles that of conventional sequence specific transcription factors shown here and in other studies (*Chen et al., 2014b*) rather than the H1.0-like slow diffusion evident in granule cells.

Although the functional consequences of these differences in the dynamic behavior of MeCP2 in neurons with different nuclear architectures remain to be investigated, our data suggest that the consequences of specific MeCP2 mutations may vary substantially between cell types (*Kron et al., 2012*; *Sugino et al., 2014*). For example, in the compact small nuclei typical of granule cells, interneurons and glia, the functions of the IDD may be dominated by its three AT-hook domains. These impart its linker histone like diffusive properties, and they may reflect dynamic roles in chromatin organization that are quite distinct from those conveyed by its stable binding to high affinity sites. In the large nuclei typical of Purkinje cells, long-range projection neurons and motor neurons, MeCP2 diffusion resembles conventional transcription factors. One might speculate that the dominant role of the IDD in these environs is to assemble protein complexes through its NID domain when stably bound to DNA. In this scenario, the relative contributions of the IDD subdomains can vary dependent on cell type, local chromatin structure, and the epigenetic microenvironment, and the phenotypic consequences of specific mutations to MeCP2 would reflect the interplay between these factors and gene expression in each specialized cell type.

## Concluding remarks

Single molecule imaging studies of transcription factors in living nuclei have uncovered critical new features of their function that could not have been predicted based on biochemical or genomic approaches (*Chen et al., 2014b*; *Chong et al., 2018*; *Hansen et al., 2018*; *Liu and Tjian, 2018*; *Loffreda et al., 2017*; *Teves et al., 2016*). Here we have presented a series of experiments that investigate in detail the molecular mechanisms that govern the dynamic behavior of MeCP2 in live neurons. Our results illustrate the complexity of MeCP2-chromatin interactions, which are sensitive to the level and state of cytosine methylation, depend on multiple protein domains, and differ depending on cell specific nuclear architecture. Although the dynamic behavior of MeCP2 resembles the linker histone H1.0, the specific features we have documented here allow MeCP2 function to be modulated locally as a result of changes in DNA methylation status and nuclear architecture. Given the variations evident in these determinants between cell type and in response to specific cues in a

given neuronal lineage, our data support a dynamic role for MeCP2 in regulation of local chromatin structure.

## Materials and methods

### Primary granule cell culture

Cerebella of P6-P8 mice were dissected in cold CMF-PBS buffer (0.13M NaCl, 4 mM KCl, 11 mM Glucose, 3.6 mM $NaH_2PO_4 \cdot H_2O$, 1.8 mM $KH_2PO_4$, 2.00 ml of 2% solution of $NaHCO_3$, 0.5 ml of 0.5% solution of phenol red, pH 7.35, 1L of $H_2O$), incubated in 37°C Trypsin-DNase (1 g/L DNase - Worthington Biochemical-, 10 g/L Trypsin -Worthington Biochemical-, 6 mM NaOH in CMF-PBS) for 5 min and triturated 10x in 1 ml RT DNase buffer (0.5 g/L DNase, 0.35% Glucose, in BME –GIBCO-) to obtain single cell suspension. Cells were pelleted by 5 min centrifugation at 1'500 rpm, resuspended in Granule Cell Medium (BME, L-Glu, Pen/Strep, 10% Horse Serum, 1% Glucose) and incubated for 1 hr at 37° C 5% $CO_2$. After recovery, cells were pelleted by 5 min centrifugation at 1'500 rpm, responded in Granule Cell Medium and plated on poly-D-lysine coated 18 mm glass cover slip. Cells were then kept in culture (at 37°C, 5% $CO_2$) for 12–14 days, exchanging half of the media with Serum-Free medium (DMEM w/o Phenol Red -GIBCO-, 25 mM HEPES, L-Glu, Pen/Strep, 1% glucose, B-27 and N2 supplement -GIBCO-) every 2 days. In case expressing vectors were inserted into the cells, following the 1 hr recovery, cells were washed 1x in RT PBS and transfected with ~15 µg of DNA plasmid using Amaxa nucleofector 2b (Lonza, program O-005). After transfection, cells were transferred in 10 ml RPMI (GIBCO) + 1% Glucose and incubated for 10 min at 37°C for recovery before being resuspended in Granule Cell Medium and plated on poly-D-lysine coated glass cover slip for culture.

### Single molecule labeling and acquisition

Culture cells were incubated with 5 nM of JF549 Halo-Ligand for 20 min and washed 3x with in Serum-Free medium. 2D single-molecule acquisition were conducted at 37°C in Serum-Free medium on a Nikon TI-Eclipse widefield Microscope equipped with a 100X Oil-immersion objective (Nikon, Plan Apo N.A. = 1.45), perfect focusing systems and DU-897 (iXon, Andor). Emission filter (Quad TIRF Set_Chroma, ZET 405/488/561/635, ET 600/50 m) was placed in front of the cameras for JF549 acquisition. We used a 561 nm laser (Agilent technologies) of excitation intensity ~100 W/cm$^2$ to achieve sparse labeling. Acquisition times were 10 ms (100 Hz) at ~100 W/cm$^2$ for mobile molecule tracks. The microscopy, lasers and the cameras were controlled through NIS-Elements (Nikon, US).

### Single molecule analysis

2D single-molecule trajectories were determined using SLIMfast/evalSPT, a Matlab-based, modified version of MTT (*Sergé et al., 2008*); 5 µm$^2$/s was set as maximum expected diffusion coefficient ($D_{max}$), which determines the maximum distance ($r_{max}$) between two frames for a particle translocation. Diffusion coefficients of individual molecules (D) were calculated from tracks with at least five consecutive frames and no more than 150 by MSDanalyzer with a minimal fitting $R^2$ of 0.8 (*Tarantino et al., 2014*). Fraction of bound molecules and diffusion coefficient of unbound molecules was determined per individual cell by using the SpotON 2-states (bound-free) kinetic model with 0.08 µm$^2$/s as maximum $D_{bound}$ and 0.15 µm$^2$/s as minimum $D_{free}$ (*Hansen et al., 2018*).

### Dwell time analysis

Sparsely labeled nuclei were acquired at low laser power (~18 W/cm$^2$) at 2 Hz. 2D single molecule localizations were determined trough SLIMfast/evalSPT with $D_{max}$ set at 0.05 µm$^2$/s. The duration of individual tracks (dwell time) was directly calculated based on the track length. This was used to calculate survival probabilities to extract the average short-lived and long-lived residence times by fitting to double exponential decay.

### Statistical analyses

Student t-test and ANOVA multipule comparison anlysese were performed using Prism (GraphPad Software, Inc). Kolmogorov-Smirnov test was performed using R Studio. For Kolmogorov-Smirnov test in panel 5C and 6B we used python 3.7 and the scipy library 0.14.

## Generation of MeCP2haloTag Knock-In animals

### Targeting vector construction

The targeting vector was constructed using the recombineering technique described in *Liu et al. (2003)*. An 8,333 bp genomic DNA fragment containing exon 3 and 4 of the *Mecp2* gene was retrieved from BAC clone RP23-77L16 to a modified pBS vector containing the DT gene, a negative selection marker. A linker-Halo-tag transgene was inserted before TGA stop codon followed by an frt-NeoR-frt cassette for ES cell selection. The length of the 5' homologous arm is 3,226 bp and that for the 3' arm is 5,095 bp.

### ES cell targeting and screening

The targeting vector was electroporated into F1 hybrid of 129S6 x C57BL/6J ES cells derived by Janelia Transgenic Facility. The G418 resistant ES clones were screened by nested PCR using primers outside the homologous arms paired with primers inside the Halo gene for 5' arm and the neo cassette for the 3' arm. The primers sequences were as follows:

5' arm screening primers: *Mecp2* Scr F1 (5'-AAAGCCAGCCTGCCTTGTTA −3') and HaloTag scr 5R1 (5'-CAACATCGACGTAGTGCATG −3'); The primers for the nested PCR are MECP Scr F2 (5'-ATGCACATGATGGGTACCTG −3') and HaloTag scr 5R2 (5'-CGATTTCAGTTGCCACTGGA −3').

3' arm screening primers: frt scr 3F1 (5'- TTCTGAGGCGGAAAGAACCA −3') and *Mecp2* scr 3R1 (5'- TGCTTTCAGTGCTAAGCAGG −3'); The primers for nested PCR are: frt scr 3F2 (5'-GGAACTTCATCAGTCAGGTAC −3') and MECP2 scr 3R2 (5'- CCAAGATGGCACTTAGTTCC-3').

### Generation of chimera and genotyping

The PCR positive ES clones were expanded for generation of chimeric mice.

The ES cells were aggregated with 8 cell embryos of CD-1 strain. The neo cassette was removed by breeding germline chimeras with ROSA26FLP1 (Jax stock#: 003946, Back crossed to C57bl/6 j for 13 generations) females.

The F1 pups were genotyped by PCR using primers *Mecp2* gt wt F (5'- GCTAAGACTCAGCCTATGGT −3'), *Mecp2* gt wt R (5'- AGGTCTTCAACCTGACTGTG-3'). The PCR products are 380 bp for the wildtype allele and 259 bp for the mutant allele. The primers for 3' end genotyping are HaloTag 3F (5'-ctactctggagatttccggt −3') and *Mecp2* gt wt R (5'-AGGTCTTCAACCTGACTGTG −3'). The PCR products is 327 bp for the mutant allele.

## Mass spec on gDNA

Genomic DNA was extracted from isolated nuclei and fully digested with DNA Degradase Plus (Zymo Research). Samples were analyzed by mass spectrometry to quantify the amount of 5mCytosine and 5hmCytosine relative to Guanosine.

## 5hmc staining

Primary cerebellar granule cells from Tet1^flox/flox; Tet2^flox/flox; Tet3^flox/flox animals were co-transfected with vectors expressing Cre and DIO-MeCP2HaloTag. After 14 days in culture on glass coverslips, cells were fixed in 4% PFA for 10 min. Samples were then incubated in 2N HCl for 30 min at 37°C, washed 3x with PBS and incubated in blocking buffer (PBS, 3% BSA, 0.01% Triton X) for 30 min at RT. Following blocking, coverslip were washed 1x in PBS and incubated with primary antibody anti-5hmC (1:500 in blocking buffer, Rb polyclonal, ActiveMotif) overnight at 4°C, washed 4x in PBST (PBS, 0.01% Triton X) and incubated with Secondary antibody (1:400 in blocking buffer, Donkey anti-Rb alxaflour 488 Jackson ImmunoResearch) and TMR-Halo-Ligand (2 nM, Promega). Cells were then washed 4x in PBST, labeled with DAPI and mounted on coverslip for acquisition with confocal microscopy.

## DNA pull-down assay

293 cells were transfected with a vector expressing MeCP2HaloTag in Nuclear Extraction Buffer containing 0.5 mM DTT and protease inhibitor, and incubated on a orbital shaker for 60 min at 4°C. Samples were centrifuged at 16'000 x g for 5 min at 4°C to collect the nuclear protein extract in the supernatant. As in *Mellén et al. (2017)*, M280-streptavidin beads (8 µL per sample) were washed once in PBS 0.1% Triton X-100, and then incubated with 200 ng of biotinylated DNA probe in 300

µL of PBS, overnight at 4˚C. Then beads were washed 2x in PBS 1% Triton X-100, 3X in Wash Buffer (0.2 mM EDTA, 20% Glycerol, 20 mM Hepes-KOH pH 7.9, 0.1 M KCl, 1 mM DTT, 1 mM protease inhibitor PMSF, 0.1% Triton X-100), and incubated with protein nuclear extract in Precipitation Buffer (0.05 mM EDTA, 5% Glycerol, 5 mM Hepes-KOH pH 7.9. 150 mM KCl, 1 mM DTT, 1 mM protease inhibitor PMSF, 0.025% Triton X-100 in PBS) for 15 min at 4˚C. Beads were then washed 5X in Wash Buffer, 1X in PBS and eluted in LDS at 95˚C for 10 min. MeCP2-HaloTag abundance in the eluted fractions was then determined by Western blot stained against HaloTag (Promega).

Synthetic probes used.

- CpG DNA probe: acgtatatacgatttacgttatacgattacgatatacgatttacgttaatacgtttacgattattacgaatt tacgtttttacgaatatacgaaatacgtttaatacgtaattacgtatattacgtatatacgatttacgaattacg.
- CpA DNA probe: gcatatatgcaatttgcattatgcaattgcaatatgcaatttgcattaatgcatttgcaattattgcaaatt tgcatttttgcaaatatgcaaaatgcatttaatgcataattgcatatattgcatatatgcaatttgcaaattgca.

## Bisulfite sequencing

Whole genome bisulfite sequencing experiments were performed using TrueMethyl-Seq reagents and workflow, following manufacturer's instructions. In each condition, genomic DNA was extracted from granule cell nuclei isolated from adult male mice cerebella. At least 1 µg of DNA per sample was sonicated using Covaris-S2 system, DNA fragments of ~200 bp were end-repaired using TruSeq DNA Sample kit as per manufacturer's instructions. 4 ng of TrueMethyl sequencing spike-in controls were added to the DNA sample prior to adapter ligation. After TruSeq DNA adapters ligation, libraries were re-purified to eliminate potential contaminating compounds using 80% Acetonitrile and TrueMethyl-Seq magnetic beads. Following bisulfite conversion, desulfonation and purification, library amplifications were performed by PCR using TruSeq DNA primers and TrueMethyl-Seq reagents. Quality of libraries was assessed using High Sensitivity D1000 ScreenTape for the 2200 TapeStation system. Libraries were sequenced through HiSeq 2000. Sequencing data were aligned to UCSC mm10 *Mus musculus* genome using bsmap aligner (v2.87) (*Xi and Li, 2009*). MOABS (*Sun et al., 2014*) methylation ratio calling module (mcall) was used to summarize the methylation levels of individual cytosines.

## Single molecules recording on acute brain slice

Adult MeCP2-HaloTag mice were decapitated under deep isoflurane anesthesia, and the brain was transferred to an ice-cold dissection solution containing (in mM): 125 NaCl, 2.5 KCl, 1.25 NaH$_2$PO$_4$, 25 NaHCO$_3$, 25 dextrose, 1.3 CaCl$_2$, 1 MgCl$_2$ (pH 7.4, oxygenated with 95% CO$_2$ and 5% O$_2$). 150-µm-thick slices of the cerebellum were sectioned using a vibrating tissue slicer (Leica VT 1200S, Leica Microsystems, Wetzlar, Germany). The slices were then transferred to a suspended mesh within an incubation chamber filled with artificial cerebrospinal fluid (ACSF), same as the solution used for dissection. After 30–60 min of recovery at 35˚C, the chamber was maintained at room temperature.

The lattice light sheet (LLS) microscope was used to perform 2D single-molecule imaging and tracking in mouse brain tissue (*Chen et al., 2014a*). In preparation of experiments, LLS scope was aligned and the imaging chamber with heating block was preheated to 37˚C. Brains tissue were labeled with Janelia Dye JF549 at the concentration of 10 nM for 15mins. After repeated wash, the brain slices were adhered to the surface of 5 mm coverslips (Warner Instruments) and placed into the imaging chamber. The sample stage was moved to find representative Granule and Purkinje cells within the slice. Single-molecule imaging was then performed by exciting a single plane with a 560 nm laser at ~1333 W/cm$^2$ at the rear aperture of the excitation objective, using Bessel beams arranged in a square lattice configuration in dithered mode. The fluorescence generated within the specimen was collected by a detection objective (CFI Apo LWD 25XW, 1.1 NA, Nikon), filtered through a 440/521/607/700 nm BrightLine quad-band bandpass filter (Semrock), and eventually recorded by an ORCA-Flash 4.0 sCMOS camera (Hamamatsu). Imaging was performed for 2000 continuous frames at 100 Hz. For dwelling time analysis we acquired 600 frames at 2 Hz under laser power ~26 W/cm$^2$.

## Nuclear isolation and FACS sorting

Nuclei isolation was performed as in *Xu et al. (2018)*. Cerebellum of adult (3–5 months old) mice was manually dissected and transfer to 5 ml of cold homogenization medium (0.25 M sucrose, 150

mM KCl, 5 mM MgCl2, 20 mM Tricine pH 7.8, 0.15 mM spermine, 0.5 mM spermidine, EDTA-free protease inhibitor cocktail, 1 mM DTT, 20 U/mL Superase-In RNase inhibitor, 40 U/mL RNasin ribonuclease inhibitor). Single nuclei suspensions were obtained by homogenization with glass dounce. Tissues were homogenized by 30 strokes of loose (A) followed by 30 strokes of tight (B) glass dounce. Nuclei were isolated by adding 50 mL of a 50% iodixanol solution (50% Iodixanol/Optiprep, 150 mM KCl, 5 mM MgCl2, 20 mM Tricine pH 7.8, 0.15 mM spermine, 0.5 mM spermidine, EDTA-free protease inhibitor cocktail, 1 mM DTT, 20 U/mL Superase-In RNase inhibitor, 40 U/mL RNasin ribonuclease inhibitor), to the homogenate and by laying the 10 ml of 25% iodixanol nuclear suspension on a 27% iodixanol cushion. Nuclei were pelleted by centrifugation 30 min, 10,000 rpm, 4°C in swinging bucket rotor (SW41) in a Beckman Coulter XL-70 ultracentrifuge. The nuclear pellet was resuspended in homogenization buffer.

FACS-sorting of Purkinjie cells was performed as in *Kriaucionis and Heintz (2009)*. The cerebellar nuclear suspension of Pcp2-TRAP animals (carrying Pcp2 BAC transgenic construct carrying IL10a-GFP fusion protein) was supplemented with DyeCycle Violet (Invirogen) to 20 μM final, as this DNA dye allows for the elimination of multiple nuclei aggregates that are recorded as a single event by the flow cytometer. Purkinje cells were identified by GFP signal and nuclei were sorted with BD FAS-CAria cell sorter using 405 nm and 488 nm excitation lasers.

# Acknowledgements

We would thank Henrik Molina and Joseph Fernandez from Rockefeller University Proteomics Resource Center for mass spectrometry analysis of genomic DNA; Allison North and Carlos Rico from Rockefeller University Bio-Imaging Resource Center for technical support; AR is supported by NCI R35 CA210043. FMP was supported by EMBO long-term fellowship. NH is an investigator of the Howard Hughes Medical Institute.

# Additional information

## Competing interests

Robert Tjian: One of the three founding funders of eLife and a member of eLife's Board of Directors. The other authors declare that no competing interests exist.

## Funding

| Funder | Grant reference number | Author |
| --- | --- | --- |
| Howard Hughes Medical Institute | | Zhe Liu<br>Nathaniel Heintz |
| National Cancer Institute | R35 CA210043 | Anjana Rao |
| EMBO | Long-term Fellowship | Francesco M Piccolo |

The funders had no role in study design, data collection and interpretation, or the decision to submit the work for publication.

## Author contributions

Francesco M Piccolo, Conceptualization, Data curation, Formal analysis, Validation, Investigation, Visualization; Zhe Liu, Resources, Software, Formal analysis, Methodology; Peng Dong, Resources, Investigation, Methodology; Ching-Lung Hsu, Resources, Investigation; Elitsa I Stoyanova, Investigation; Anjana Rao, Resources; Robert Tjian, Conceptualization; Nathaniel Heintz, Conceptualization, Supervision, Funding acquisition

## Author ORCIDs

Zhe Liu (iD) http://orcid.org/0000-0002-3592-3150
Elitsa I Stoyanova (iD) http://orcid.org/0000-0001-6400-6119

Robert Tjian (iD) http://orcid.org/0000-0003-0539-8217
Nathaniel Heintz (iD) https://orcid.org/0000-0002-8874-8704

## Ethics

Animal experimentation: This study was performed in strict accordance with the recommendations in the Guide for the Care and Use of Laboratory Animals of the National Institutes of Health. All of the animals were handled according to approved institutional animal care and use committee (IACUC) protocols (# 16944) of the Rockefeller University. The protocol was approved by the Committee on the Ethics of Animal Experiments of the Rockefeller University (OLAW assurance # #A3081-01).

## Decision letter and Author response

Decision letter https://doi.org/10.7554/eLife.51449.sa1
Author response https://doi.org/10.7554/eLife.51449.sa2

# Additional files

## Supplementary files

• Transparent reporting form

## Data availability

All data generated or analysed during this study are included in the manuscript and supporting files.

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
