## [Decision Letter]

**Acceptance summary:**

The manuscript of Piccolo et al. studies the dynamics of Methyl-CpG-binding-protein in live neurons. Single-particle tracking of MeCP2 shows a mobile and immobile population, where the mobile population diffuses slower than gene specific transcription factors. The authors characterize in detail the contribution of DNA binding domain, the intrinsically disordered domain, and the AT-hook domain in the diffusion and DNA binding of MeCP2. In addition, the results show that the slow diffusion depends on non-specific chromatin interactions, and appear to be cell-type specific.

The presented experiments are well-performed and the results are convincing. The work shows new features of the complex behavior of MeCP2 and the authors present data to show how DNA methylation, chromatin compaction and specific protein sub-domains influence specific aspects of this behavior.

**Decision letter after peer review:**

Thank you for submitting your article "MeCP2 nuclear dynamics in live neurons results from low and high affinity chromatin interactions" for consideration by *eLife*. Your article has been reviewed by three peer reviewers, including Job Dekker as the Reviewing Editor and Reviewer #1, and the evaluation has been overseen by Huda Zoghbi as the Senior Editor.

The reviewers have discussed the reviews with one another and the Reviewing Editor has drafted this decision to help you prepare a revised submission.

Essential revisions:

1) This is critical point: SpotOn is a key tool that the authors use to evaluate and interpret their data. As far as I understand SpotOn assumes only one bound and one unbound state. Are the assumptions of SpotOn correct for a DNA binding factor with more than one binding state, as seems the case for MeCP2?

2) Possibly related to the previous point: MeCP2 shows very different diffusion behavior from other factors such as histones or Sox2. This also makes it more challenging to determine the individual bound and non-bound populations. For Sox2, the diffusion plot shows a bimodal population, but the MeCP2, the distribution is much broader and almost unimodal. In the mutants the distributions shifts, but it does not appear that the two populations are really separate. The bound fraction, and diffusion of bound molecules is then calculated as if these are really different populations. Could it be that this is just a single population, with binding sites of different affinity resulting in a broader distribution? Do the authors have any additional evidence to assume two different populations for MeCP2?

3) The authors state: "The rate of diffusion of the unbound R133C protein is measurably increased". Figure 3D does not show that this effect of 133C on diffusion rate is statistically significant. Is it significant or not? If not then the authors cannot make this statement.

4) Figure 5D: The number of observations in heterochromatin is much larger. When down-sampled to the same level as observed in the euchromatic domains, is the difference still significant?

5) Is diffusion of MeCP2 in Purkinje cells still limited by the AT-hook domains? If not, then other interpretations for the altered mobility than altered chromatin compaction could explain the results.

6) Since transfection efficiency can change between experiments, and between cells, it would be nice if the authors could show that the concentration of transfected MeCP2 is similar to the endogenous level. If no antibody is available to perform Western blots, the authors could perform FCS on the HaloTagged cells to determine the concentration in endogenously-tagged and transfected cells. This is especially important for all the mutated constructs where the mutations and truncations could affect protein level or stability, and lower concentration may affect the fraction bound.

7) It somewhat disappointing that the difference in nuclear dynamics of MeCP2 in granule versus Purkinje cell was not examined in more detailed. For example, how similar are the affects of the Rett mutations on MeCP2 nuclear dynamics in these two types neurons?

---

## [Author Response]

Essential revisions:1) This is critical point: SpotOn is a key tool that the authors use to evaluate and interpret their data. As far as I understand SpotOn assumes only one bound and one unbound state. Are the assumptions of SpotOn correct for a DNA binding factor with more than one binding state, as seems the case for MeCP2?

We have now rewritten the first section of the Results to clarify this issue. In brief, there are two different types of imaging experiments that are used to reveal the dynamic properties of each protein included in the study.

The first is to acquire images at low excitation and long acquisition times (~2Hz) to assess the residence times of stably bound molecules (Figure 1—figure supplement 1). The dwelling times in our study match those previously determined for Sox2, TBP and histone H2B in ES cells. In agreement with those studies, the dissociation kinetics of MeCP2 is best described by a double exponential decay that accounts for short-lived and long-lived populations of bound molecules (Figure 1—figure supplement 1B) for all proteins studied, including MeCP2 and histone H1.0. Furthermore, the short-lived bound molecules of MeCP2 have dwell times of ~ 2 seconds, similar to the sequence specific transcription factors Sox2 and TBP, and histone H1.0.

The second technique is to track individual molecules at high speed (100Hz) to reveal rapid diffusion dynamics. This allows one to assess the fraction of bound molecules because they diffuse very slowly (<0.08μm^2^/s). However, long-lived and short-lived stable binding events cannot be distinguished in these experiments, and they are pooled into one number represented by the fraction bound. For the majority of data we have presented, we have used SpotOn to analyze fast tracking particles which we are able to monitor for no longer than ~2 seconds (200 frames at 10ms/frame) because the molecules are quickly bleached at high laser powers. SpotOn is used to model two states (bound and unbound) in each cell acquired at these high rates of imaging. It enables us to determine the fraction of MeCP2 that is stably bound, and the rapid rate of diffusion of unbound MeCP2 molecules that are imaged at these high speeds.

2) Possibly related to the previous point: MeCP2 shows very different diffusion behavior from other factors such as histones or Sox2. This also makes it more challenging to determine the individual bound and non-bound populations. For Sox2, the diffusion plot shows a bimodal population, but the MeCP2, the distribution is much broader and almost unimodal. In the mutants the distributions shifts, but it does not appear that the two populations are really separate. The bound fraction, and diffusion of bound molecules is then calculated as if these are really different populations. Could it be that this is just a single population, with binding sites of different affinity resulting in a broader distribution? Do the authors have any additional evidence to assume two different populations for MeCP2?

Studies from several laboratories (Nan et al., 1993; Mellen et al., 2012; Gabel et al., 2015) have demonstrated that high affinity binding of MeCP2 to methylated DNA (Kd ~ 1x10^-9^M) is similar to the binding of sequence specific transcription factors to their cognate binding sites. As discussed above, the residence time data we have presented confirms this fact and demonstrates that the short-lived stable interactions of MeCP2, Histone H1.0, Sox2 and TBP are similar. Based on these data, and previous studies of Sox2 and TBP in ES cells (Chen et al., 2014b), a diffusion coefficient of <0.08 μm^2^/s was chosen to calculate the fraction of stably bound MeCP2.

The conclusion that the unbound fraction of MeCP2 is not engaged in stable binding interactions with high affinity binding sites (5mCpG, 5mCpH, 5hmCpH) is supported by the fact that its average diffusion (~1μm^2^/s) is more an order of magnitude faster than the values demonstrated for high affinity transcription factor interactions, and by our structure/function studies indicating that domains of MeCP2 outside of the MBD can impact its diffusion in the presence of a mutation (R106W) that completely abolishes stable binding.

The plots of diffusion coefficients shown in Figure 1E appear mono-modal because the diffusion of many more of the unbound MeCP2 molecules is slower than that measured for TBP or Sox2 despite the fact they are not stably bound to high affinity sites (note this is a logarithmic scale). Changes in the shapes of these plots are informative, and they almost certainly reflect a range of low affinity interactions with DNA or proteins that are mediated by the AT-Hook and NID domains. SpotOn analyzes unbound molecules as a single population to provide an average diffusion coefficient so that one can compare quantitatively the behavior of the entire unbound population.

3) The authors state: "The rate of diffusion of the unbound R133C protein is measurably increased". Figure 3D does not show that this effect of 133C on diffusion rate is statistically significant. Is it significant or not? If not then the authors cannot make this statement.

We have now modified the manuscript to reflect the reviewers’ comments. Although, as indicated by the reviewer, the difference in the average diffusion of MeCP2 R133C per cell is not statistically significant (Figure 3D), the distribution of diffusion coefficients is noticeably shifted and there is a significant trend toward higher diffusion coefficients (Figure 3C). We think it is important to point this out to the reader, and we have done so in very conditional terms. R133C results in increased diffusion coefficient, but this is a very subtle effect that is not measured by SpotOn fitting model over all the cells analyzed (Figure 3D).

4) Figure 5D: The number of observations in heterochromatin is much larger. When down-sampled to the same level as observed in the euchromatic domains, is the difference still significant?

To understand whether this might alter the conclusions we have reached, we have performed additional statistical tests. Thus, we sub-sampled euchromatic data to match the number of particles recorded in heterochromatin and calculate the p value with Kolmogorov–Smirnov test from data presented in Figure 5C (all cells) and in 5D (individual representative cells). In order to minimize data loss, we repeated this analysis in 1000 iterations of randomly selected particle values from the pool of euchromatin. In Author response image 1 we plotted the log10 distribution of the p-values calculated. These data nicely confirm the statistic reported on panel 5D and indicate that the difference is not due to a sampling bias (The red dashed lines represent p-value=0.05).

5) Is diffusion of MeCP2 in Purkinje cells still limited by the AT-hook domains? If not, then other interpretations for the altered mobility than altered chromatin compaction could explain the results.

While it would be very nice to test this directly, transfection of mutant constructs into Purkinje cells in slice preparations is exceedingly inefficient, and the technical difficulties associated with imaging very sparsely expressing cells in these preparations have prevented structure/function studies of this type. However, we believe the very different dynamic behaviors of MeCP2 molecules in these two very different nuclear types are fascinating and should be presented.

Of course, the implications of our data are many and significant. As alluded to by the reviewers, they certainly suggest that the IDD domain of MeCP2 (including the AT Hooks) has very little effect on diffusion in Purkinje cells and other nuclei of this type, and that this difference may have important functional consequences. Rather than enumerate these possibilities and speculate on the causes of this behavior (higher levels of 5hmC, differences in 3D chromatin architecture, altered levels of histone H1, etc..), we chose to make a very general statement that we believe is consistent with the data we have presented. If there are other interpretations of the data or the reviewer would like to suggest different language, we could edit the text appropriately.

6) Since transfection efficiency can change between experiments, and between cells, it would be nice if the authors could show that the concentration of transfected MeCP2 is similar to the endogenous level. If no antibody is available to perform Western blots, the authors could perform FCS on the HaloTagged cells to determine the concentration in endogenously-tagged and transfected cells. This is especially important for all the mutated constructs where the mutations and truncations could affect protein level or stability, and lower concentration may affect the fraction bound.

To address this concern, in Author response image 2 we calculated total nuclear fluorescence of endogenous expressing MeCP2 HaloTag as well as the transfected HaloTag constructs used in Figure 4 and labelled with HaloLigand JF549 in representative cells. Nuclear fluorescence of each representative cell was normalized to background levels outside the nucleus of each cell. As expected, transfected samples display lower signal/noise ratios than endogenous expressed MeCP2HaloTag, and the levels measured for each construct are similar.

**Author response image 2. respfig2:** 

Additionally, in Figure 1 and extensively in Figure 1—figure supplement 3 we show that endogenously and exogenously expressed HaloTag constructs behave identically indicating that levels of MeCP2 expression do not impact on its dynamic behavior in our analysis.

7) It somewhat disappointing that the difference in nuclear dynamics of MeCP2 in granule versus Purkinje cell was not examined in more detailed. For example, how similar are the affects of the Rett mutations on MeCP2 nuclear dynamics in these two types neurons?

See response to point 5. This analysis is beyond the scope of this paper given existing technology for exogenous expression in Purkinje cells.